# Experimental Evaluation of the Role of Ecologically-Relevant Hosts and Vectors in Japanese Encephalitis Virus Genotype Displacement

**DOI:** 10.3390/v11010032

**Published:** 2019-01-06

**Authors:** Ajit K. Karna, Richard A. Bowen

**Affiliations:** 1Department of Microbiology, Immunology, and Pathology, Colorado State University, Fort Collins, CO 80523, USA; ajit.karna@gmail.com; 2Department of Biomedical Sciences, Colorado State University, Fort Collins, CO 80523, USA

**Keywords:** Japanese encephalitis virus, duck, *Culex quinquefasciatus*, genotype displacement, host competence, vector competence, Asia

## Abstract

Japanese encephalitis virus (JEV) is a flavivirus that is maintained via transmission between *Culex* spp. mosquitoes and water birds across a large swath of southern Asia and northern Australia. Currently JEV is the leading cause of vaccine-preventable encephalitis in humans in Asia. Five genotypes of JEV (G-I–G-V) have been responsible for historical and current outbreaks in endemic regions, and G-I and G-III co-circulate throughout Southern Asia. While G-III has historically been the dominant genotype worldwide, G-I has gradually but steadily displaced G-III. The objective of this study was to better understand the phenomenon of genotype displacement for JEV by evaluating both avian host and mosquito vector susceptibilities to infection with representatives from both G-I and G-III. Since ducks and *Culex quinquefasciatus* mosquitoes are prevalent avian hosts and vectors perpetuating JEV transmission in JE endemic areas, experimental evaluation of virus replication in these species was considered to approximate the natural conditions necessary for studying the role of host, vectors and viral fitness in the JEV genotype displacement context. We evaluated viremia in ducklings infected with G-I and G-III, and did not detect differences in magnitude or duration of viremia. Testing the same viruses in mosquitoes revealed that the rates of infection, dissemination and transmission were higher in virus strains belonging to G-I than G-III, and that the extrinsic incubation period was shorter for the G-I strains. These data suggest that the characteristics of JEV infection of mosquitoes but not of ducklings, may have play a role in genotype displacement.

## 1. Introduction

Japanese encephalitis virus (JEV) belongs to the family *Flaviviridae* and genus *Flavivirus*, and is an enveloped, single-stranded positive-sense RNA virus having a genome approximately 11 kb in size [1,2]. JEV exists in an enzootic transmission cycle in which the infectious Culicine mosquito, primarily *Culex tritaeniorhynchus*, transmits the virus via bite to ardeid birds such as herons and egrets (reservoir hosts) and pigs (amplifying host) [3,4]. The mosquito-bird and the mosquito-pig cycles are independent of each other, and humans are incidental hosts in either transmission cycle [3,4,5]. This zoonotic virus is neuroinvasive and neurovirulent in humans and, although it causes substantial morbidity among all unvaccinated age groups, mortality is seen predominantly in children [6,7]. JEV is the leading cause of arboviral encephalitis in the world and is common in regions where paddy fields, pig husbandry, and mosquitoes co-exist [6,8]. With an estimated three billion people living in currently known at-risk areas globally, the challenge remains in non-JEV endemic areas where the known JEV-vectors have been reported. Moreover, the possibility exists for introduction of JEV into areas in Africa, Europe, and the Middle East where populations of competent mosquito vectors have been reported [6,8,9,10]. An effective vaccine for use in humans is commercially available, yet 50,000–60,000 cases of Japanese encephalitis (JE) are reported annually worldwide, largely because of the shortfalls in vaccine supply and the cost of vaccine itself [4,6,11,12].

Japanese encephalitis virus is thought to have emerged from ancestral viruses during the 19th century in the region of the Malay [6]. Historical descriptions of human illness having clinical manifestations compatible with JE suggest infections occurring as early as 1871 [6,12]. The first definitive report of JEV outbreaks dates to 1924, when an outbreak occurred in Japan [6,13]. Five genetic subtypes of JEV, referred to as genotypes, have been defined based on phylogenetic analysis of nucleotide sequence of the envelope protein gene and designated G-I through G-V [14,15,16]. Viruses from each genotype have been responsible for historical and current outbreaks in humans and other animals in endemic regions since its emergence [6,12,13,17]. JEV of different genotypes can be distinguished antigenically [7,18,19], and it is thought that effective cross-genotype immune protection is conferred by infection with viruses belonging to anyone of G-I, G-II, G-III or G-IV viruses [7]. At present, there is no data on cross-genotype protection information for G-V. Although currently available vaccines based on G-III protect against G-I infection, the protection may be sub-optimal [7,20,21,22,23], suggesting that displacement of G-III by G-I may adversely affect current vaccine effectiveness. The occurrence of genomic variation between genotypes has been thought to enable one genotype of JEV to become more fit than the other one in the ecological niches in which they circulate [9,13,24,25,26,27]. Competitive displacement among JEV genotypes has been previously observed including complete or partial displacement of G-III by G-I strains in Asia [27,28,29,30,31] and G-II by G-I strains in Australia [9,13,24,25,26,28]. G-III strains were dominant between 1935 and 1990, but G-I strains began to appear in Indonesia, Thailand and Cambodia in the 1970s and gradually became dominant in most parts of Asia. Although differences in virulence and neuro-invasiveness were not identified [7,31], G-I dominated over other JEV genotypes in Japan, Korea, Thailand, Taiwan and Vietnam [30,31]. Co-circulation and genotype/strain displacement are common phenomenon among flaviviruses. It was observed with West Nile virus (WNV), a closely related flavivirus, when the WNV-02 genotype displaced the WNV-NY99 genotype in the USA [32,33]. Among other well-studied examples, the endemic American genotype of dengue virus type-2 (DENV-2) was displaced by the Southeast Asian genotype of DENV-2 [34] and a native DENV-3 strain was displaced by an invasive DENV-3 strain in Sri Lanka in the 1980s [35]. Similarly, the DENV-2 Asian-American genotype NI-1 clade was replaced by the NI-2B clade in Nicaragua [36]. Such displacements have several potential impacts on the international public health infrastructure, including the possibility that currently available vaccines are sub-optimal and that future displacement events could perhaps jeopardize current vaccination programs.

The mechanisms responsible for dominance of G-I strains and displacement of G-III strains remains poorly understood. Studies based on nucleotide sequence analysis have described genetic diversity, evolutionary history, and host composition over the course of genotype displacement [13,26,37]. Studies based on surveillance of vectors and hosts have demonstrated frequent introductions of G-I strains in areas previously dominated by G-III strains [25,27,28,38,39]. A study by Schuh et al., [28] used cultured cell-based (*Aedes albopictus* C6/36 cells and duck embryo fibroblasts) approaches to explain the JEV genotype displacement and found that JEV G-I isolates indicated significantly higher infectivity titer than JEV G-III isolates at 24, 36, and 48-hour post-infection in C6/36 cells. Another study tested susceptibilities of *Cx. quinquefasciatus* to JEV G-I and G-III strains, and found North American *Cx. quinquefasciatus* as a competent vector for JEV G-I and G-III strains but JEV G-III demonstrated significantly higher infection rate than JEV G-I and non-demonstrable difference in transmission rate between both genotypes [40]. However, this latter study [40] incorporated only a single isolate of G-III and two isolates of G-I, limiting the inferences that could be drawn and did not address infection in vertebrate hosts. What has been lacking in previous attempts to explain JEV genotype displacement is evaluation of ecologically relevant avian hosts and mosquito vectors with the same strains of virus. Due to continued evolution, JEV remains as an emerging worldwide human health threat and disease severity within the human population could be increased by an emergence of new genotypes with altered disease manifestations, transmission potential, or resistance to current vaccines. The study presented here used hosts and the vectors involved in natural JEV ecology, and was designed to enhance our understanding of JEV genotype displacement with the hypothesis that JEV G-I has an enhanced ability to replicate in mosquitoes and/or avian hosts relative to JEV G-III.

## 2. Materials and Methods

### 2.1. Ethics Statement

The study was approved by the Animal Care and Use Committee of Colorado State University (protocol number: 16-6477A) and was conducted in biosafety level-3 facility approved by the Association for the Assessment and Accreditation of Laboratory Animal Care, International (AAALAC).

### 2.2. Cells and Viruses

Vero (African green monkey kidney epithelial) cells, a mammalian cell type that does not possess functional interferon signaling, obtained from the American Type Culture Collection (Manassas, VA, USA) were maintained in Dulbecco’s modified Eagle’s medium (DMEM) containing 5% bovine calf serum (BCS), penicillin (100 U/mL) and streptomycin (50 μg/mL) at 37 °C with 5% CO_2_. Stock viruses were inoculated in Vero cells and fresh viruses were harvested at the end of 4 days post-inoculation (dpi). The strains of JEV utilized in this study are described in Table 1. All virus strains were titrated by double-overlay plaque assay in Vero cells using techniques previously described [41].

### 2.3. *In Vitro* Growth Curve Kinetics

Five replicates of a multi-step growth curve in Vero cells were obtained for all six viruses using a multiplicity of infection (MOI) of 0.01. Briefly, after inoculating the cells with viruses, the plates were incubated for 2 h at 37 °C with 5% CO_2_, virus inoculum was removed, cells were washed once with PBS and 5 mL fresh growth media (DMEM with 5% FBS) was added. Samples for virus titration were collected on the day of inoculation (0-day post-inoculation, dpi), and daily for 8 days; at each sampling, 0.5 mL of supernatant was harvested and replaced with the same amount of fresh growth media. The supernatants were supplemented with 0.1 mL FBS and stored at −80 °C until assayed by plaque assay in Vero cells.

### 2.4. Animal Experiments

Indian runner ducks (*Anas platyrhynchos domesticus* obtained from Murray McMurray Hatchery, Webster City, IA, USA) were housed under biosafety level-3 containment. Thirty ducklings were randomly allocated to six groups, corresponding to the six virus strains to be tested. Ducklings were banded with their specific numbers assigned to them. One extra duckling was added to either groups inoculated with MAR864 and Sagiyama strains, but was not inoculated and served as contact control. The duckling-cages were kept away from each other to prevent any contacts between groups. The ducklings were fed standard waterfowl feed and had water available *ad libitum*. The ducklings were inoculated subcutaneously with ~10^6^ PFU of each virus at 5 to 6 days of age. A 100 μL blood sample was withdrawn from each duckling daily and immediately diluted into 450 μL of BA-1 medium (DMEM containing 1% bovine serum albumin, 250 mg/L sodium bicarbonate, gentamicin (50 mg/L), penicillin (100,000 IU/L) and streptomycin (50 mg/L)) yielding a serum dilution of approximately 1:10. Body temperature and weight were measured three times at a single time-point and average measurement was recorded daily from the day before inoculation through 7 dpi and on 10, 14, and 21 dpi. Oral and cloacal swab samples were taken on 3, 4 and 5 dpi and immediately added to 450 μL of BA-1 medium supplemented with amphotericin B (2.5 mg/L), polymyxin B (50,000 U/L). and twice the standard concentration of penicillin, streptomycin, gentamicin. The samples were immediately stored at −80 °C until assay. All ducklings were euthanized on 21 dpi.

### 2.5. Mosquito Experiments

*Culex quinquefasciatus* Say mosquitoes were obtained from a colony maintained at the Arthropod-borne and Infectious Diseases Laboratory at Colorado State University. This colony was established using mosquitoes collected in Sebring County, Florida in 1988. Larvae of the mosquitoes were provided with powdered fish food (Walmart, Fort Collins, CO, USA), and when pupae started emerging, they were transferred in a cup with water inside a container covered with a mesh. The insectary was maintained at 26–27 °C, 16:8 light: dark cycle, and 70–80% relative humidity. Female mosquitoes of the F11–F13 generations (5–7 days post-eclosion) were used in this study.

### 2.6. Vector Competence and Extrinsic Incubation Period

Vero cells cultured in 25 cm^2^ flasks were inoculated with each of the six strains of JEV G–I and G-III at an MOI of 0.01. Four days later, the supernatant from each flask was collected, centrifuged (10,000× *g* for 8 min at 4 °C) to remove cellular debris, and the fresh, clarified supernatants were used for feeding mosquitoes. Mosquitoes were fed with defibrinated cattle blood mixed with freshly harvested viruses at a 1:1 ratio that was titrated at the time of use by plaque assay in Vero cells. The blood-virus mixture was pipetted into the loading chamber of a Hemotek membrane feeding system (Hemotek, UK) covered with pork casing (Whole Foods Market, Fort Collins, CO, USA), and the mosquitoes were allowed to feed for 1 h at ambient room condition at 26 °C and 80% relative humidity. Blood-engorged mosquitoes for each virus strain were briefly cold anesthetized (4 °C) and separated into two groups for assay at day 7 and 14 post-feeding (dpf). The blood-engorged mosquitoes were incubated at 26 °C and 70% relative humidity with *ad libitum* water and sugar cubes. On 7 and 14 dpf, 60 mosquitoes per virus strain were cold anesthetized (4 °C), and legs and wings were removed and transferred into a clean tube containing two stainless steel beads and 250 μL mosquito diluent (PBS supplemented with 20% heat-inactivated FBS and antibiotics (100 mg/L gentamicin, 200 IU/L penicillin G, 100 mg/L streptomycin, and 5 mg/L amphotericin B)). The mosquito bodies were placed with their proboscis inserted in a capillary tube containing immersion oil (about 5 μL) for at least 30 min. After salivation, mosquito bodies were transferred into a separate tube with mosquito diluent and beads. The ends of capillary tubes containing immersion oil and saliva were broken off into micro centrifuge tubes containing 100 μL mosquito diluent. Mosquito tissues (bodies, and legs + wings) were homogenized at 25 cycles/second for 1 min in a mixer mill and subsequently centrifuged at 15,000× *g* for 3 min at 4 °C. To determine the infection status of mosquito tissues and saliva, 50 μL of each sample was inoculated onto Vero cell monolayers in two replicate wells of a 96-well plate and incubated at 37 °C. After incubation for an hour, 150 μL of DMEM supplemented with 10% FBS was added into each well and the plates were incubated for 24 h at 37 °C. Media from the 96-well plates inoculated with mosquito samples was decanted at 24 h; cells were washed once with PBS, fixed with 70% acetone for 1 h and dried. The cells were stained by addition of 50 μL of diluted mouse anti-JEV hyper-immune ascitic fluid (Source: Centers for Disease Control and Prevention, Fort Collins) (1:100 in PBS) for 1 h at 37 °C, washed in PBS, then incubated for 1 h with 50 μL of DyLight conjugate of Goat anti-Mouse Ig (H + L, 1:100 in PBS) (ThermoFisher Scientific, Waltham, MA, USA). The plates were washed again with PBS and observed under a fluorescent microscope. The sample was determined positive if replicate wells demonstrated fluorescence and negative if none of the wells demonstrate fluorescence. Controls in each assay consisted of duplicate wells inoculated with BA-1 diluent or BA-1 spiked with stock JEV. The infection rate was determined as the proportion of mosquito bodies that were found infected to the total number of mosquitoes exposed to the infectious blood meal. Dissemination rate was determined as the proportion of mosquito legs and wings that were found infected to the total number of mosquitoes exposed to the infectious blood meal. Similarly, transmission rate was determined as the proportion of mosquito saliva samples that were found infected to the total number of mosquitos exposed to the infectious blood meal.

### 2.7. Virus Titration and Serology

Replication of JEV in ducklings was measured as viremia titer. Plaques were counted on one and two days after the second overlay, and viremia titers were expressed as plaque forming units (PFU)/mL. The minimum concentration of JEV in serum or swab samples that could be detected using this assay was set at ~100 PFU/mL. Neutralizing antibody titers in sera were determined by plaque reduction neutralization test (PRNT). Briefly, 2-fold serial dilutions (starting at 1:10) of heat-inactivated (56 °C for 30 min) duckling sera were mixed with equal volumes of stock virus (JEV VN strain, genotype I) diluted to ~200 PFU/mL. After 1 h of incubation, the serum-virus mixtures were inoculated into wells in 6-well plates of confluent Vero cells and processed subsequently as described for plaque assay. Plaques were counted one day after the second overlay and neutralization titer was expressed as the reciprocal of the highest dilution of the serum that inhibited ≥90% of JEV plaques (PRNT_90_).

### 2.8. Data Analysis

Virus titer in Vero cells culture and viremia titer in ducklings were expressed as mean (±1 SE). The mean peak virus titer of each of the six JEV strains inoculated in Vero cells were compared using one-way ANOVA, and the time x strain interaction was analyzed by repeated measures ANOVA. Similarly, the mean peak viremia titers of the six strains in ducklings were compared using one-way ANOVA. Furthermore, infection and dissemination percentages were arcsine-square root transformed to render them normal, normality was checked on transformed data and compared by standard least squares method between G-I and G-III in the mosquito study. Since the data on transmission proportions cannot be normally distributed, as 50% of the values are tied 0’s, transmission proportions for G-I on 7 dpf, G-III on 7 dpf, G-I on 14 dpf and G-III on 14 dpf were pooled and compared by Fisher’s exact test. SAS 9.4 (SAS Institute, Cary, NC, USA) was used for all statistical analysis. Graphs were made on GraphPad Prism 7.0 (La Jolla, CA, USA). The statistical significance was declared at *p* < 0.05.

## 3. Results

### 3.1. *In Vitro* Growth Kinetics

The growth kinetics of all six strains of JEV were comparable in Vero cells (Figure 1). Five of the viruses reached a peak titer on 2 dpi (mean ± 1 SE, log_10_PFU/mL): KE-093-83 (8.6 ± 0.3), MAR864 (8.5 ± 0.2), JE-91 (8.8 ± 0.1), CH392 (9.0 ± 0.04), and Sagiyama (8.7 ± 0.2). The JKT27-087 virus achieved peak titer on 3 dpi (8.1 ± 0.3). When comparing only the mean peak virus titer (2 dpi), one-way ANOVA revealed a significant difference in the peak virus titer among strains (F_5_ = 8.9, *p* <0.0001), and Tukey-Kramer post-hoc analysis indicated that the mean peak virus titer of JKT27-087 differed significantly from the mean peak virus titers of the rest of the strains (*p* < 0.05). Similarly, the mean peak virus titers of G-I (8.6 ± 0.1) and G-III (8.4 ± 0.2) were not significantly different (one-way ANOVA, F_1_ = 1.1, *p* = 0.3). Repeated measures ANOVA revealed a significant effect of strain on virus titer (F_5_ = 2.7, *p* = 0.02) and a significant time x strain interaction among six strains (F_40_ = 2.4, *p* < 0.001), and post-hoc Tukey-Kramer multiple comparison revealed only CH392 strain significantly different than JKT27-087 strain (*p* = 0.009). Virus titer for all strains declined after 3 dpi, corresponding to the observation of cytopathic effect in the Vero cells.

### 3.2. Clinical Signs in Ducklings Inoculated with JEV

None of the ducklings inoculated with JEV showed apparent signs of disease or distress. A mild elevation of body temperature (preinoculation body temperature range was 40.0–42.8 °C) on 1 dpi was observed in ducklings inoculated with MAR864, CH392, JKT27-087, and Sagiyama strains, however general time effects on body temperature of virus-inoculated ducklings were hard to discern (Figure 2). The weight of ducklings increased linearly from about 100 g on 0 dpi to approximately 700 g in 22 days, and it demonstrates that the virus infection in ducklings did not reduce the daily gain in weight with the increase in age (Figure 3). Both the contact-control ducklings remained healthy and gained weight similar to the virus-inoculated ducklings.

### 3.3. Replication of JEV in Ducklings

The back-titrated dose of virus inoculated into ducklings ranged from 5.5 to 6.0 log_10_PFU/mL. The viremia titer with individual strains (Figure 4) showed considerable consistency among the strains and between the genotypes in (i) duration to reach peak viremia titers, (ii) the peak titers, and (iii) the time viable virus is not detected in the serum. For all six strains and both genotypes, peak viremia was observed 2–3 dpi, and the mean peak viremia titers (mean ± 1 SE, log_10_PFU/mL) were: KE-093-83 (4.1 ± 0.2), MAR864 (3.3 ± 0.2), JE-91 (4.2 ± 0.1), CH392 (3.0 ± 0.8), JKT27-087 (4.2 ± 0.3), and Sagiyama (3.3 ± 0.6). Similarly, the mean peak viremia titer (mean ± 1 SE, log_10_PFU/mL) in the ducklings for G-I and G-III were 3.9 ± 0.2 and 3.5 ± 0.3, respectively. The mean peak viremia titers of ducklings were neither significantly different among the six strains (oneway ANOVA, F_5_ =1.6, *p* = 0.2), nor were they significantly different between genotypes (Student’s *t*-test, t_28_ = −1.02, *p* = 0.3). Virus was not detected after 3 dpi in any of the ducklings, nor in either of the two non-inoculated control ducks.

### 3.4. Shedding of JEV from Ducklings

Virus shedding in oral and cloacal secretions at and above the limit of detection (2 log_10_PFU/mL) was observed on 3, 4, and 5 dpi (Figure 5 and Figure 6). All 5 ducklings inoculated with KE-093-83 (3.4 ± 0.2, log_10_PFU/mL, mean ± 1 SE), JE-91 (2.5 ± 0.2) and JKT27-087 (3.9 ± 0.3), 4 of 5 ducklings inoculated with MAR864 (2.9 ± 0.2) and Sagiyama (3.7 ± 0.1), and 2 of 5 ducklings inoculated with CH392 (3.6 ± 0.2) shed viruses orally on 3 dpi. On 4 dpi, 1 out of 5 ducklings inoculated with KE-093-83 (2.0), JE-91 (3.2), and CH392 (3.5); 2 out 5 ducklings inoculated with Sagiyama (3.2 ± 0.7); and 4 out 5 ducklings inoculated with MAR864 (4.0 ± 0.3) and JKT27-087 (3.3 ± 0.3) shed viruses orally. On 5 dpi, only a single duckling from KE-093-83, MAR864 and JKT27-087 shed viruses orally of titer 2.5, 3.1, and 3.4 log_10_PFU/mL, respectively.

Regarding cloacal shedding of virus strains from ducklings, 1 out of 5 ducklings inoculated with CH392 (3.2), 2 out of 5 ducklings inoculated with KE-093-83 (3.4 ± 0.9) and MAR864 (2.2 ± 0.2), 3 out of 5 ducklings inoculated with JE-91(2.5 ± 0.3) and JKT27-087 (3.6 ± 0.1), and 4 out of 5 ducklings in Sagiyama (3.9 ± 0.2) shed viruses through cloaca on 3 dpi. On 4 dpi, cloacal shedding of virus was observed in ducklings inoculated with KE-093-83 only, and that 2 out of 5 ducklings shed the virus at a titer of 3.4 ± 0.7 log_10_ PFU/mL. On 5 dpi, 1 out of 5 ducklings inoculated with MAR864 (2.8), JE-91 (2.3), and JKT27-087 (3.3), and 2 out of 5 ducklings inoculated with KE-093-83 (2.3 ± 0.3) shed viruses cloacally.

### 3.5. Antibody Response to JEV in Ducklings

Each duckling was seronegative (PRNT_90_ titer <10) before inoculation with the viruses as demonstrated by the absence of virus-specific antibodies. All ducklings that were inoculated with virus, except one in the CH392 group, seroconverted by 7 dpi, and the neutralizing antibody titer (PRNT_90_) increased by 21 dpi (Table 2). Neither of the non-inoculated contact-control ducklings developed a detectable antibody response by 21 dpi, indicating a lack of contact transmission.

### 3.6. Vector Competence and Extrinsic Incubation Period

To test the hypothesis that G-I displaced G-III because it is more infectious for *Cx. quinquefasciatus* than G-III, competence of *Cx. quinquefasciatus* for JEV G-I and G-III viruses was measured among 60 mosquitoes for each virus strain. The concentration of virus fed to the mosquitoes for each virus strain ranged from 5.1–5.9 log_10_PFU/mL (Table 3). *Cx. quinquefasciatus* mosquitoes were found competent to transmit all six strains of JEV, but competence varied between JEV G-I and JEV G-III strains (Table 4). Interestingly, at 7 dpf, only G-I strains were detected in mosquito saliva (see Table 4 for % of mosquitoes in which JEV was detected) but not G-III strains as measured by immunofluorescence based-plaque assay in Vero cells, indicating a longer extrinsic incubation period (EIP) of JEV G-III (at least more than 7 days) than JEV G-I (as late as 7 days) in *Cx. quinquefasciatus*. By 14 dpf, all the six virus strains were detected in saliva. The findings supported our hypothesis, although dissemination percentage was not significantly different (t_4_ = −1.7, *p* = 0.20), infection percentage (t_4_ = −4.8, *p* = 0.009) and transmission percentage (2-tailed *p* = 0.0004) were significantly different between G-I and G-III on 7 dpf in *Cx. quinquefasciatus*. However, on 14 dpf, infection percentage (t_4_ = −1.3, *p* = 0.30), dissemination percentage (t_4_ = −2.0, *p* = 0.10), and transmission percentage (2-tailed *p* = 0.051) were not significantly different between G-I and G-III. Although, the effect of genotype x dpf on infection percentage (F_3_ = 2.5, *p* = 0.1) and dissemination percentage (F_3_ = 2.5, *p* = 0.1) was not significant, the effect was significant on transmission percentages (F_3_ = 9.5, *p* = 0.005). Our findings demonstrated JEV G-I is relatively better at infecting mosquitoes, disseminating through body parts of mosquitoes and being shed through saliva at an earlier time point.

## 4. Discussion

Avian hosts are critical sources of JEV in endemic areas [5,41,42], and field data suggest ducks are infected by JEV [43,44]. Previous studies involving experimental infection in ducklings with JEV suggest that young ducks develop high magnitude viremia, compatible with serving as reservoir hosts [41]. Furthermore, it has been reported that domestic ducks infected with JEV can infect biting *Culex* species mosquitoes [45]. In the current study, we observed growth of virus in vitro and in vivo with all six strains of the JEV, three of each belonging to either JEV G-I or JEV G-III. All six strains of JEV reached peak titer of approximately 8–9 log_10_PFU/mL on 2–3 dpi in Vero cells. Ducklings inoculated with freshly-grown viruses did not show signs of disease, but viremia was detected in serum obtained from all except 3 ducklings (two in CH392 group, one in Sagiyama group) on 1 to 3 dpi. Viremia ranged from 2–5.5 log_10_PFU/mL; previous studies indicated that only 10 PFU of JEV is enough to infect mosquitoes [46,47]. There was no significant difference on the peak viremia titer among the six viruses tested in ducklings. Both JEV G-I and JEV G-III have been demonstrated to have similar pathogenic potential in mouse models, which is higher than representative strains of the other JEV genotypes [7]. Overall, we were not able to demonstrate a difference in fitness between viruses from G-I versus G-III based on replication in ducks, suggesting that differences in the ability to replicate in avian hosts is not the basis of genotype displacement.

All but one duckling seroconverted by 7 dpi and had higher antibody titer on 21 dpi, regardless of inoculated strain of JEV. Consistent with the demonstration that JEV G-III strain vaccine elicits higher antibody titer than JEV G-I strain vaccine in mouse model [7], our results indicate substantial seroconversion rate after inoculation with JEV G-III compared to JEV G-I on 7 dpi. JEV was also detected in oral and cloacal swabs obtained from ducklings inoculated with JEV G-I and JEV G-III, as has also been reported by Ricklin et al. [23] in pigs with the detection of RNA of JEV. Only infected ducklings demonstrated oral and/or cloacal shedding of JEV. Little is known about the significance, if any, of oral or cloacal shedding in the transmission of JEV. Although the current study was not specifically designed to test contact transmission, the lack of transmission to co-housed ducklings suggests that JEV is unlikely to be transmitted through the oral route in ducks.

*Cx. quinquefasciatus* mosquitoes are known to be competent vectors for JEV in several areas of Asia [5,48], and the current study further confirmed the competence of this species for JEV. We observed higher infection, dissemination and salivary transmission with G-I than G-III. We did not measure infection, dissemination and transmission titer. Our study is consistent with higher infectivity previously reported for G-I than G-III for cultured mosquito cells [28] and *Cx. quinquefasciatus* in vivo [40]. JEV G-I also disseminated more rapidly than JEV G-III in *Cx. quinquefasciatus*. Since, JEV infection in Vero cells inoculated with 7 dpi saliva content of mosquitoes which were blood-fed with JEV G-III viruses was not detected, our study suggests a difference in extrinsic incubation period in mosquitoes as a responsible factor in genotype displacement. A shorter extrinsic incubation period of WNV 02 genotype strains of West Nile virus was reported to be the cause of WNV NY99 genotype displacement in the US [32,33]. Genotype displacement of WNV in the US [32,33], DENV-2 in the US [34], strain displacement in DENV-3 in Sri Lanka [35], clade displacement of DENV-2 in Nicaragua [36] demonstrated differences in replicative efficiency and transmission potential among circulating virus genotypes, strains, or clades, which is also applicable to JEV genotype displacement.

In addition to differences in EIP and host competence, other factors that may have contributed to genotype displacement include: (i) adaptation to the specific virus variants of mosquito species prevalent in different regions of Asia, (ii) the introduction of modernized farming practices including management of rice paddy fields and isolation of pig farms from urban environments, (iii) the implementation of concerted JEV immunization programs across many Asian countries, and (iv) the possible impact of climate change on migratory bird patterns and mosquito distribution. Although these represent possible mechanisms for displacement, evidence for those mechanisms are lacking.

Several caveats to the current study must be acknowledged. First, we utilized the Indian runner ducks in our study for several reasons: (i) they are a common domesticated breed of duck throughout Asia [49,50,51,52] and in endemic zones of JEV [17,53], (ii) previous studies from our laboratory confirmed them to be susceptible hosts for JEV, and (iii) ducklings have been detected with anti-JEV antibodies in JEV-endemic regions [45,54,55]. Second, we utilized *Cx. quinquefasciatus* instead of known principle vector of JEV, *Cx. tritaeniorhynchus*, due to unavailability of the latter vector in the US. The *Cx. quinquefasciatus* used in this study have been maintained in a colony for several decades, and we understand vector competence for flaviviruses is impacted by a long-term colonization of mosquito vectors [56]. However, previous studies that utilized short-term colonized *Cx. quinquefasciatus* [40] nonetheless showed a comparable vector competence, suggesting that our findings may reflect the actual dynamics of JEV in *Cx. quinquefasciatus*.

Future studies should include *Culex tritaeniorhynchus* from areas where displacement has been recorded, and evaluate vectorial capacity, extrinsic incubation period and virus titers in mosquitoes with single and mixed genotype infection. However, such mixed infections of JEV G-III and JEV G-I may not occur at a high rate (proportionally) in the field environment. In other words, competition may be demonstrated in the laboratory, but may be insignificant in the field since the frequency of mosquitoes being co-infected by both genotypes may not be sufficiently high to influence the predominant genotype. Host age is another feature that should be evaluated. The young ducklings used in the current study likely were not fully immunocompetent and differences among viral genotypes may have been masked at that young age.

The possibility of potential impact of the concerted immunization campaigns against JEV which may have occurred when G-III was pre-eminent, that is, G-III would have been affected by the vaccine leaving a void for G-I to fill. In this context, the timeframe in phylogenetic terms (temporal predictions for emergence/divergence) should be considered in future studies. It is possible that the presence of G-III as the predominant genotype for many years, coupled with vaccination using a G-III-based vaccine (SA-14-14-2) resulted in a high level of immunity to G-III in the community that was suboptimal in cross-protecting against G-I strains, facilitating establishment of G-I strains [57].

The broader impact of our study is that exploration of host-virus interaction by the utilization of ecologically relevant hosts and vectors will elucidate the mechanistic basis behind genotype emergence and displacement. Viral and host ecology are key components of disease emergence and spread, and this finding can extend into an exploration of other zoonotic viruses that co-circulate, co-evolve and show displacement events. Our study substantially lends empirical support to the potential importance of variation in EIP as an explanation for genotype displacement in JEV, and will allow theoreticians to formulate a mathematical model to explore the impact of this variation at a population level. Moreover, the widespread and close proximity of domestic ducks and humans throughout Asia suggests that if ducks indeed serve as an important reservoir for JEV, they may have an impact similar to pigs in amplifying the virus. Clearly, a one health approach to control of this pathogen will be critical to formulating effective control strategies.

## Figures and Tables

**Figure 1 viruses-11-00032-f001:**
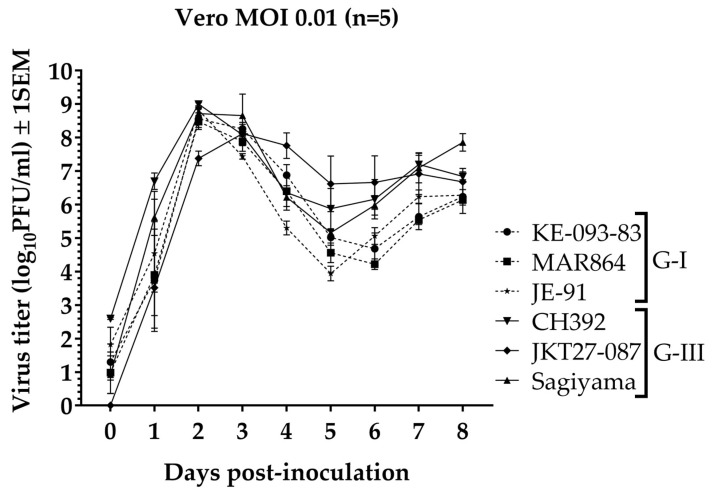
Growth curve of Japanese encephalitis virus G-I and G-III strains in Vero cells.

**Figure 2 viruses-11-00032-f002:**
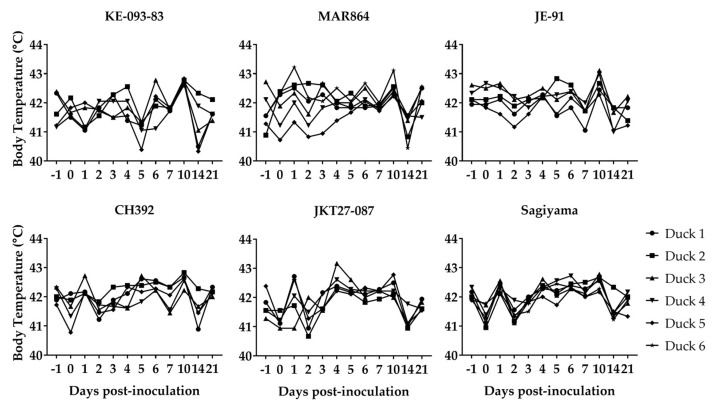
Body temperature in ducklings inoculated with Japanese encephalitis viruses. Duckling 6 for both MAR864 and Sagiyama groups were non-inoculated controls.

**Figure 3 viruses-11-00032-f003:**
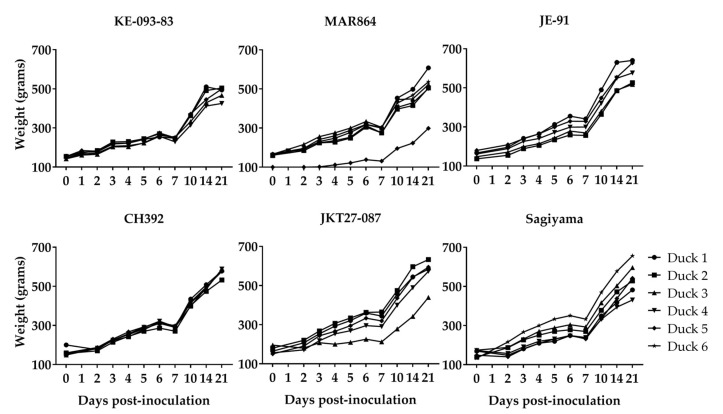
Body weight of ducklings inoculated with Japanese encephalitis viruses. Duckling 6 for both MAR864 and Sagiyama groups were non-inoculated controls.

**Figure 4 viruses-11-00032-f004:**
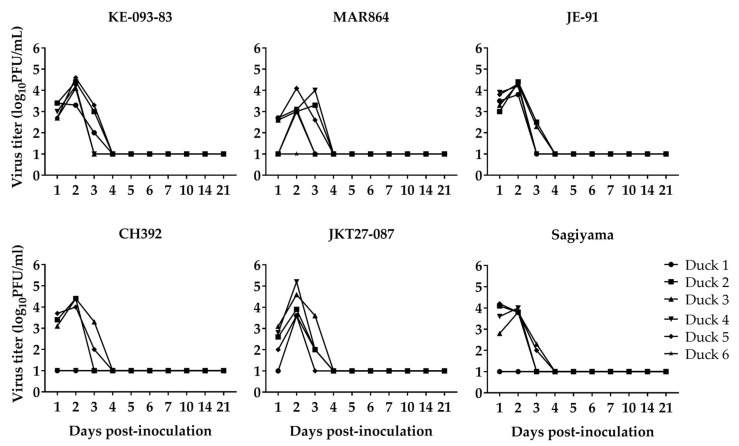
Viremia in ducklings inoculated with Japanese encephalitis viruses. Ducking 6 for both MAR864 and Sagiyama groups were non-inoculated controls. Virus titers < 2 were below the limit of detection.

**Figure 5 viruses-11-00032-f005:**
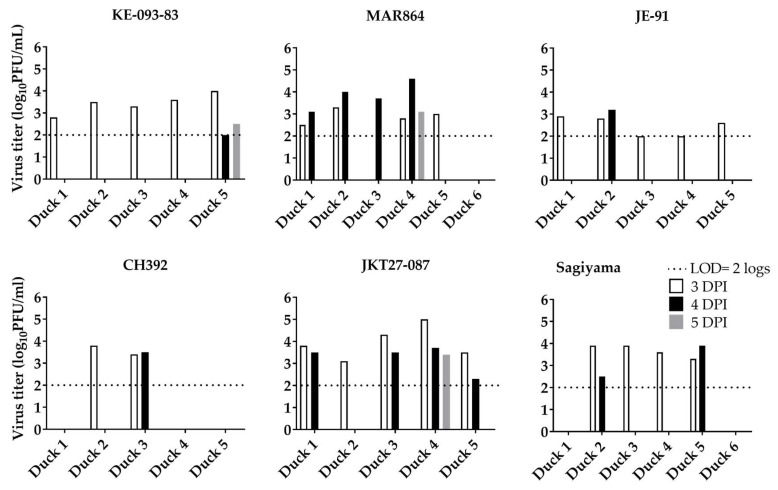
Oral shedding of Japanese encephalitis viruses from ducklings. Duckling 6 for both MAR864 and Sagiyama groups were non-inoculated controls.

**Figure 6 viruses-11-00032-f006:**
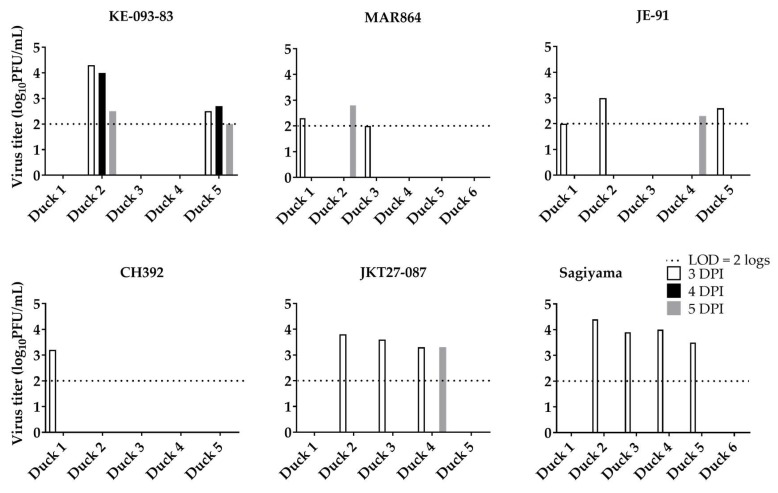
Cloacal shedding of Japanese encephalitis viruses from ducklings. Duckling 6 for both MAR864 and Sagiyama groups were non-inoculated controls.

**Table 1 viruses-11-00032-t001:** Japanese encephalitis viruses utilized in the study.

Virus Strain (Genotype)	Source	Collection Year	Passage History	Country (Climate)	GenBank Accession #
KE-093-83 (G-I)	Mosquito	1983	Vero#1	Thailand (Tropical)	KF192510
MAR864 (G-I)	*Cx. tritaeniorhynchus*	1967	C6/36#1, Vero#1	Cambodia (Tropical)	D00983
JE-91 (G-I)	*Cx. tritaeniorhynchus*	1991	C6/36#1, Vero#1	Korea (Temperate)	GQ415355
CH392 (G-III)	*Cx. tritaeniorhynchus*	1987	Vero#1, C6/36#2	Taiwan (Sub-tropical)	U44961
JKT27-087 (GIII)	Mosquito	1987	C6/36#1	Indonesia (Tropical)	JQ429308
Sagiyama (G-III)	*Cx. tritaeniorhynchus*	1957	C6/36#1	Japan (Temperate)	D00972

**Table 2 viruses-11-00032-t002:** Plaque Reduction Neutralization Test antibody titer in ducklings inoculated with Japanese encephalitis viruses (90% cut off).

G-I					G-III				
Virus	Duck	0 dpi	7 dpi	21 dpi	Virus	Duck	0 dpi	7 dpi	21 dpi
KE-093-83	D1	<10	40	320	CH391	D1	<10	<10	80
	D2	<10	20	320		D2	<10	40	160
	D3	<10	40	≥640		D3	<10	40	≥320
	D4	<10	20	80		D4	<10	160	160
	D5	<10	80	320		D5	<10	40	80
MAR864	D1	<10	160	160	JKT27-087	D1	<10	40	≥320
	D2	<10	40	80		D2	<10	≥320	≥320
	D3	<10	20	160		D3	<10	80	≥320
	D4	<10	10	160		D4	<10	20	≥320
	D5	<10	20	160		D5	<10	40	80
Control	D6	<10	<10	<10	Sagiyama	D1	<10	40	80
JE-91	D1	<10	40	160		D2	<10	20	≥320
	D2	<10	80	80		D3	<10	80	≥320
	D3	<10	80	320		D4	<10	160	80
	D4	<10	80	80		D5	<10	40	160
	D5	<10	80	160	Control	D6	<10	<10	<10

**Table 3 viruses-11-00032-t003:** Titration of the freshly harvested and the blood meal Japanese encephalitis virus strains.

Strain	Genotype	Type	PFU/mL	Log_10_PFU/mL
KE-093-83	I	Fresh harvest	4,100,000	6.6
		Blood meal (1:1)	750,000	5.9
MAR864	I	Fresh harvest	1,300,000	6.1
		Blood meal (1:1)	390,000	5.6
JE-91	I	Fresh harvest	1,400,000	6.1
		Blood meal (1:1)	320,000	5.5
CH392	III	Fresh harvest	100,000	6.0
		Blood meal (1:1)	350,000	5.5
JKT27-087	III	Fresh harvest	450,000	5.7
		Blood meal (1:1)	140,000	5.1
Sagiyama	III	Fresh harvest	1,200,000	6.1
		Blood meal (1:1)	1,000,000	6.0

**Table 4 viruses-11-00032-t004:** Rates of infection, dissemination, transmission at 7- and 14-days post-feeding infectious blood meal to mosquitoes.

Virus strain	Genotype	DPF	Infection (%)	Dissemination (%)	Transmission (%)
KE-093-83	G-I	7	56/60 (93.3%)	18/60 (30%)	1/60 (1.6%)
		14	47/60 (78.3%)	33/60 (55%)	4/60 (6.6%)
MAR864	G-I	7	56/60 (93.3%)	21/60 (35%)	8/60 (13.3%)
		14	49/60 (81.6%)	11/60 (18.3%)	5/60 (8.3%)
JE-91	G-I	7	47/60 (78.3%)	26/60 (43.3%)	3/60 (5%)
		14	58/60 (96.6%)	43/60 (71.6%)	11/60 (18.3%)
CH392	G-III	7	24/60 (40%)	15/60 (25%)	0/60 (0%)
		14	13/60 (21.6%)	10/60 (16.6%)	3/60 (5%)
JKT27-087	G-III	7	35/60 (58.3%)	5/60 (8.3%)	0/60 (0%)
		14	42/60 (70%)	7/60 (11.6%)	2/60 (3.3%)
Sagiyama	G-III	7	32/60 (53.3%)	19/60 (31.6%)	0/60 (0%)
		14	51/60 (85%)	13/60 (21.6%)	4/60 (6.6%)

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
