# Peer review of "Experimental Evaluation of the Role of Ecologically-Relevant Hosts and Vectors in Japanese Encephalitis Virus Genotype Displacement"

_viruses, 2019, doi:10.3390/v11010032_

Reviewer 1 Report

In this manuscript, the authors use relevant avian and mosquito models to examine genotype displacement of Japanese encephalitis virus (JEV). They should be commended for working with ducklings, a natural vertebrate host, which is not straight forward in a BSL-3, and conducting mosquito studies with large numbers per group (n=60 per condition). They work with three low passage JEV isolates from two different genotypes (G-I and G-III). Multistep growth curves in Vero cells were not significantly different in peak titers for the six isolates. Viremia and shedding in ducklings were also not significantly different. In contrast, G-I isolates had greater infection and transmission rates at 7 days post-feeding, resulting in a shorter extrinsic incubation period in mosquitoes, compared to G-III isolates. This result is a possible explanation for genotype displacement. There are some concerns as described below.

Concerns:

1.      In Figures 2-6 with the animal data, the legend should make it clear which animals are control animals (duck #6 for MAR864 and Sagiyama groups). Also, in Figure 4, it appears that some of the inoculated ducks did not develop a viremia (e.g. ducks #1 and #4 in CH392 group). This should be pointed out when the results are discussed. Also, line 348 states that all ducks developed viremia, which the data in Figure 4 does not support.

2.      There appears to be a correlation between no detectable viremia and no detectable shedding (e.g. duck 6 inoculated with MAR864). If there is an association, this should be discussed.

3.      Line 305 (discussion of Table 2) and line 356 – it states that all virus-inoculated ducks had seroconverted by day 21, but duck1 inoculated with CH391 had not seroconverted by day 7.

4.      Ducklings at 5-6 days of age are not fully immunocompetent. Thus, it is possible that differences between viral genotypes may be masked at this young age. In other words, an older duck may demonstrate differences in viremia with the different genotypes. While it is outside the scope of this manuscript to repeat these studies in older ducks (and difficult to conduct in a BSL-3 laboratory), the authors should discuss this potential issue in the Discussion.

Author Response

We thank the reviewer for excellent comments.

1) We modified the caption for Figures 2-6 to indicate which ducklings were controls.

2) Another good observation which we incorporated into the Discussion.

3) Corrected as suggested.

4) Another good suggestion that we incorporated into the Discussion.

Reviewer 2 Report

A sound manuscript. However, there are some inconsistencies between the text describing results and the graphs.

Lines 51, 52

use more original references if possible

Lines 135..

why were control animals not added to other groups?

Line 142

why was serum ‘roughly’ diluted? Easy enough to add 900 µL medium to 100 µL sample?

Line 150

were these mosquitoes analysed for potential co-inhabiting insect-specific viruses?

Line 226

JK peaks on day 3 (Fig 1), re-calculate following data if necessary

Line 245 and Fig 2

could authors please convert °F to °C (or at least add °C in brackets)?

Line 253

fig 2 Legend: add ‘non-infected control’ (or similar) to ‘Duck 6’, same for Fig 3

Line 269

add ‘they’: ..nor were they significantly…

Line 271

fig 4 It is not clear from text if there is detectable viremia after day 4? Graphs show all ducklings, incl controls, with detectable virus. Should this line be on ‘0’?

in graphs ‘MAR 864’ and’ Sagiyama’, are the control ducklings infected?

in ‘CH392’ graph, 2 ducklings show no peak in viremia – this should be described in text (paragraph line 260 etc)

in ‘Sagiyama’ graph, it is difficult to discern pattern but it seems that duckling 1 is also not showing peak in viremia (full circles for days 1 and 2)

Line 280

fig 5 shows 2 ducklings, infected with CH392 genotype, shedding virus orally on day 3.  This needs to be added to text

add ‘non-infected control’ to ‘Duck 6’

Line 307

‘contract’ should be ‘contact’

Line 390

insert ‘was’: …in which JEV was detected…

Line 342

Previous studies… suggest, not suggests

Line 346

JK in Fig 1 shows peak titre at day 3

Line 348

….viremia was detected in serum…from all ducklings on 1 to 3 dpi: see comments above regards Fig 4

Line 349               

viremia ranged from 1 to 5.5… see Fig 4

10 PFU/mL

Line 359

insert ‘compared to’ instead of ‘than’

Line 380

this paragraph should be expanded (i.e. give examples) since it is the main topic

Line 389

sentence after ii) needs to be re-written

Line 390

delete ‘ducks are’

Line 392

comma after bracket. Their habitat overlaps with …. Rather than waterfowls overlap

Line 396

instead of ‘The Cx. quinquefasciatus utilized’, use ‘Cx. quinquefasciatus mosquitoes used in this

study’ have been …

delete ‘more than’

Lines 401-407    

maybe vectorial capacity can be used rather than vector competence in your future study?

Line 409               

choose another word for ‘hit’

Lines 411-416    

this long, convoluted sentence should be re-written to make a clear statement

Author Response

Excellent comments - thank you.

Lines 51, 52 use more original references if possible: We considered this but retained the view that these two review papers provide the best summary of that history and we would otherwise have to cite numerous original sources.

Lines 135.. why were control animals not added to other groups?  We received an extra two ducklings in the shipment and decided to incorporate them as controls in two groups; manuscript not changed.

Line 142 why was serum ‘roughly’ diluted? Easy enough to add 900 µL medium to 100 µL sample?  Changed 'roughly' to 'approximate2ly'.  This is because we did not measure the hematocrit on each sample to determine whether exactly what fraction of the sample was serum versus cells, but assumed 50% of the 100 ul was serum.

Line 150 were these mosquitoes analysed for potential co-inhabiting insect-specific viruses?  No we did not do that.

Line 226 JK peaks on day 3 (Fig 1), re-calculate following data if necessary.  Edited to reflect that only 5 of 6 peaked on day 3.

Line 245 and Fig 2 could authors please convert °F to °C (or at least add °C in brackets)?  Conversion performed as suggested.

Line 253 fig 2 Legend: add ‘non-infected control’ (or similar) to ‘Duck 6’, same for Fig 3: Modified as suggested

Line 269 add ‘they’: ..nor were they significantly…  Done

Line 271 fig 4 It is not clear from text if there is detectable viremia after day 4? Graphs show all ducklings, incl controls, with detectable virus. Should this line be on ‘0’? The limit of detection was 2 logs/ml; we modified the text to indicate that viremia was not observed past day 3 in any duckling and inserted a not regarding limit of detection in the legend of Fig 4.

in graphs ‘MAR 864’ and’ Sagiyama’, are the control ducklings infected?  No, they were not infected; text edited to indicate that finding.

in ‘CH392’ graph, 2 ducklings show no peak in viremia – this should be described in text (paragraph line 260 etc) - indicated such in discussion, where it seemed to fit better.

in ‘Sagiyama’ graph, it is difficult to discern pattern but it seems that duckling 1 is also not showing peak in viremia (full circles for days 1 and 2):  correct - modified discussion

Line 280 fig 5 shows 2 ducklings, infected with CH392 genotype, shedding virus orally on day 3.  This needs to be added to text

add ‘non-infected control’ to ‘Duck 6’

Line 307 ‘contract’ should be ‘contact’: corrected!

Line 390 insert ‘was’: …in which JEV was detected: corrected

Line 342 Previous studies… suggest, not suggests: corrected

Line 346 JK in Fig 1 shows peak titre at day 3: corrected

Line 348 viremia was detected in serum…from all ducklings on 1 to 3 dpi: see comments above regards Fig 4: corrected

Line 349  viremia ranged from 1 to 5.5… see Fig 4: the limit of detection for viremia was 2.0; no change made

Line 359 insert ‘compared to’ instead of ‘than’: corrected

Line 380 this paragraph should be expanded (i.e. give examples) since it is the main topic.  We are unaware of concrete evidence (examples) for those mechanisms, but edited the paragraph to indicate so.

Line 389 sentence after ii) needs to be re-written:  yes indeed!

Line 390 delete ‘ducks are’: done

Line 392 comma after bracket. Their habitat overlaps with …. Rather than waterfowls overlap: re-written

Line 396 instead of ‘The Cx. quinquefasciatus utilized’, use ‘Cx. quinquefasciatus mosquitoes used in this study’ have been … delete ‘more than’: revised as suggested.

Lines 401-407 maybe vectorial capacity can be used rather than vector competence in your future study?  Good suggestion - added

Line 409 choose another word for ‘hit’: done

Lines 411-416 this long, convoluted sentence should be re-written to make a clear statement:  re-written as suggested.

Reviewer 3 Report

Revue_ Viruses_40912

Experimental Evaluation of the Role of Ecologically-Relevant Hosts and Vectors in Japanese Encephalitis Virus Genotype Displacement

Ajit Karna and Richard Bowen explored the behaviors of JEV viruses representative of genotype I and III within their duck natural reservoir and a potential vector ‘Culex quinquefasciatus”

Exploring the two systems is really an hard work that deserved to be nicely described and it was..

The article is very well writen and easy to follow, both the objective and scientific litterature within the introduction is very nicely described and provide a very good overview of the displacement of various arbovirus genotype.

The material and method is complete and all reported data and analysis are clearly supported by the appropriate statistical methods. The data clearly shown no differences between the viral genotype in the reservoir hosts while the extrinsic incubation period (EIP) in mosquitoes clearly differentiate genotype I and III.

The discussion is also very well conducted and all relevants information were provided.

However, Iin my mind the EIP difference should be also reflected within the kinetics of epidemics related to the two genotypes. Did the authors analyzed the litterature according this hypothesis?

I have only few minor comments:

The analysis was performed with quite old strain of JEV (isolated between 1957 to 1991) that is probably before the displacement observed and described in the introduction by articles from the 2010’s. Could the authors comment this point? About the GIII strain, the group of Vanlandingham use the GIII Tairan strain that seems to be as efficient as the GI KE-93-93. Please comment

Minor remarks:

1-    please provide the body temperature using the international unit (ie °Celcius) both in the text (line 245 and in the figure 2 scale).

2-    Within this figure 1 and the following fig 3 and 4 please indicate clearly the mock-infected duckling control (or non-inoculated ??)/ within the MAR864 and Saglyama group. This is not obviously clear from the material and method that these are only in two groups out of 6.

3-    Within the discussion line 399 about the vector competence of short-term adapted mosquitoes versus old laboratory strain please provide some numbers and if possible statistical analysis.

4-    At the end of the article in the concluding sentences please use the term extrinsic incubation period in full.

Author Response

Thank you for your time and useful comments.  No, we were not able to find dagta to allow analysis of EIP differences relative to kinetics of epidemics caused by the two genotypes.  

Regarding the isolation dates of the strains of JEV we evaluated: G-III viruses were dominant prior to 1990, but displacement appears to have begin during the 1970's, and the isolation dates of the strains we evaluated span the general time period during which displacement was beginning to occur.

The excellent report by Dr. Vanlandingham's group did demonstrate similar competence of Cx. quinquefasciatis mosquites using the Tairan strain, but we are not albe to explain that in comparison to the results we obtained with a battery of other strains; such analysis may well require comparison of full genome sequences, which was not conducted.

1 please provide the body temperature using the international unit (ie °Celcius) both in the text (line 245 and in the figure 2 scale). - We converted all temperatures to C 

2 Within this figure 1 and the following fig 3 and 4 please indicate clearly the mock-infected duckling control (or non-inoculated ??)/ within the MAR864 and Saglyama group. This is not obviously clear from the material and method that these are only in two groups out of 6. - Added text to the figure legends as indicated.

3 Within the discussion line 399 about the vector competence of short-term adapted mosquitoes versus old laboratory strain please provide some numbers and if possible statistical analysis. Apologies, but we do not have such numbers.

4 At the end of the article in the concluding sentences please use the term extrinsic incubation period in full.  Done